# The Role of Surgery for Oligometastatic Non-Small Cell Lung Cancer

**DOI:** 10.3390/cancers14102524

**Published:** 2022-05-20

**Authors:** Caleb J. Euhus, Taylor R. Ripley, Cristian G. Medina

**Affiliations:** The Michael E. DeBakey Department of Surgery, Division of Thoracic Surgery, Baylor College of Medicine, 1 Moursund St., Houston, TX 77030, USA

**Keywords:** oligometastatic non-small cell lung cancer, metastatic non-small cell lung cancer, NSCLC history, lung cancer staging, surgery and NSCLC, local consolidative therapy (LCT), locally-aggressive therapy (LAT)

## Abstract

**Simple Summary:**

This article clarifies the context and definition of the term oligometastasis as it applies to non-small cell lung cancer and reviews the current results in the use of surgery for its management. Ideally after reading this article, practitioners will be better equipped to pick patients for locally aggressive therapy, and researchers will be primed to design the next generation of studies that prioritize treatments in the management of this historically deadly disease.

**Abstract:**

Oligometastatic non-small cell lung cancer (NSCLC) is metastatic disease that refers to a limited number of metastatic sites. It is analogous to an intermediate stage of NSCLC, between localized and widely metastatic disease, even though no staging criteria establishes this distinction. Oligometastatic NSCLC describes a patient subgroup with limited metastasis to one or a few organs. These patients seem to have a more indolent cancer than those with diffuse metastasis. For these select patients with oligometastatic disease, the use of palliative systemic therapy over local aggressive treatment may be a missed opportunity to improve survival. The clear definition of this subgroup and identification of the best treatment remains the current challenge in the management of the disease. Surgery was the early cornerstone in the treatment of limited disease; however, as modalities such as chemotherapy, stereotactic radiosurgery, and immunotherapy have matured, the role of excision is less clearly defined. There are sparse randomized controlled trials comparing the efficacy of different treatment modalities in patients with oligometastatic NSCLC. However, there is a growing body of retrospective research detailing the prognostic factors that characterize the role of surgery in the management of these patients. This article clarifies the context and definition of the term oligometastatic, as it applies to NSCLC, and reviews the current results in the use of surgery for its management.

## 1. Introduction

Lung cancer accounts for 28% of all cancer mortality in the United States, inflicting 160,000 deaths annually [1,2]. Approximately 80% of newly diagnosed lung cancers are classified as NSCLC, and 60–70% of these are advanced at diagnosis [3]. Previously, patients with metastatic NSCLC were not considered to be candidates for curative-intent treatment, due to their poor prognosis, with a median survival of 8–11 months and 5-year overall survival (OS) rate of only 4–6% [4]. Platinum-based chemotherapy extends survival from the early reports of 3–4 months to 12 months. First generation targeted therapy offers similar results [5]. Patients with advanced NSCLC harboring the rearrangement of the anaplastic lymphoma kinase (ALK) gene can receive crizotinib [6], and the median progression free survival (PFS) with this drug is only 10.9 months. Alectinib, a second-generation ALK inhibitor, was found to be more efficacious, extending PFS to 34.8 months [7]. Other first-line agents target the epidermal growth factor receptor (EGFR) and ROS proto-oncogene 1 (ROS1). All of these modern targeted therapies offer a PFS between two and three years. There is now growing optimism that improved targeted therapies will allow a larger cohort of patients the chance to benefit from locally aggressive treatment (LAT) with surgery or radiotherapy. However, in spite of these advances in systemic options, truly long-term survival is still relatively rare. Additionally, the management of patients with oligometastatic disease versus metastatic disease is less defined.

Management of oligometastatic NSCLC should be considered separately, compared to diffuse metastatic disease. There is mounting evidence of significantly improved OS in patients with oligometastatic disease who undergo LAT with either surgery or radiotherapy, in addition to systemic therapy. However, which patients will benefit from surgery or radiotherapy (or a combination of the two) is less clear. Additionally, the precise selection criterion for the number or size of the metastatic lesions are uncertain.

### 1.1. Defining Oligometastatic Disease in NSCLC

The evolution of the surgical treatment of lung oligometastatic disease stems from early surgeon optimism, after successful outcomes with aggressive treatment of multiple primary lung tumors. In 1975, Martini and Melamed published a case series of 50 patients with two or three primary carcinomas of the lung [8]. These patients underwent resection of second primary lung cancers. Based on this case series, with reasonable outcomes, they concluded that having multiple tumors is not an exclusion criteria for surgical candidacy. Additionally, they noted that survival in patients with multiple primary carcinomas of the lung was similar to those with single tumors, when those tumors were also treated by excision. This report defined the criteria for second primary lung cancer, but the outcomes of these patients sparked an interest in resection of oligometastatic lesions. In 1995, Hellman and Weichselbalm were the first to use the term oligometastasis [9]. They argued that the existing theory that cancer spreads via local or hematogenous mechanism did not adequately explain the isolated metastases that occurred in one or a few locations or organs [6]. They used the term oligometastatic disease to describe a more fastidious step in cancer progression, wherein the tumor biology limits the distant spread to a limited number of deposits in the major organs. In other words, they believed that not all cancer diffusely disseminates once it progresses beyond the primary site. Instead, certain primary tumors create metastatic tumors that behave more like satellite lesions than diffuse disease. Based on the perception that those with a handful of metastases have a less aggressive biology than those with more disseminated disease, Hellman and Weichselbalm championed a curative approach to treatment, rather than a palliative therapeutic strategy for the patients with oligometastatic disease.

Nomenclature describing oligometastasis has expanded and is more precise, compared to its establishment in 1995 [10]. First, the descriptions are based on time, relative to the primary lesions. Synchronous metastatic lesions are diagnosed at the same time as the primary tumor. In contrast, metachronous metastatic lesions occur if diagnosed after the discovery of the primary lesion. These lesions are usually diagnosed on surveillance imaging after the management of the primary lesion has been completed. In addition to a time-based descriptor, the location of the lesions has been incorporated into the staging system. The 8th edition of the TNM staging system for NSCLC, released in 2016, split metastatic disease into subcategories of M (metastatic) disease [11,12]. With this revision, M1b denotes a single extra-thoracic metastatic focus, whereas M1c denotes multiple extra-thoracic metastases. This distinction was noted based on a review of the outcomes of patients with metastatic disease, which revealed that the patient’s prognosis with a single extra-thoracic lesion was better than their prognosis with multiple extra-thoracic metastatic sites. Of note, the patient’s initial stage of disease, based on the TNM classification, remains their stage throughout the course of their disease, regardless of whether metastatic or recurrent disease occurs. That is important for oligometastatic disease, because this disease only qualifies for an M descriptor if the lesions are synchronous metastatic foci. If they are metachronous lesions, then the patient remains with the original stage and is noted to have metastatic disease at the point when these lesions appear. Attempts to increase the granularity in the characterization of oligometastatic NSCLC have been developed. Guckenberger used a systematic literature review and Delphi consensus process to develop an oligometastatic disease classification system [9]. He created a decision tree of five binary disease characterization factors, in order to propose a dynamic oligometastatic state model. Consensus in nomenclature is valuable for designing clinical trials and comparing treatment modalities for subcategories of metastatic disease.

In conjunction with the progress made in the classification of oligometastatic NSCLC, numerous studies have outlined the key prognostic factors that help to identify patients for whom LAT, such as surgery and radiation, will be beneficial. As will be discussed later in this review, younger patients, smaller tumors, metachronous disease, and a lack of lymph node involvement are associated with the best prognoses.

### 1.2. Surgery for Oligometastatic NSCLC

#### Prospective Phase II and Randomized Controlled Trials

A few prospective controlled trials, several meta-analyses, and many retrospective reviews have helped to define the prognostic indicators that clarify the role of surgery in the multimodal treatment of oligometastatic NSCLC. The early controlled trials for the management of oligometastatic disease focused on the treatment of single metastasis to the brain. These studies were not specific to NSCLC, but they did include patients with metastatic NSCLC. Additionally, the brain is a common site for metastatic spread of NSCLC; therefore, the studies are worth discussing in NSCLC reports. For patients with single metastasis to the brain and limited systemic disease, a randomized trial by Vecht and colleagues demonstrated a median survival of 7 months with whole-brain radiotherapy (WBRT) and 12 months with WBRT combined with surgical resection [13]. Patchell and colleagues extended these findings by demonstrating, in a randomized controlled trial, that excision with postoperative WBRT was superior to excision alone [14]. He found that 46% of patients treated with surgery alone experienced a recurrence in the resection site, but those treated with surgery followed by adjuvant radiation had a recurrence rate of only 10%. For the isolated brain metastasis, surgery and radiation should be considered as both independent options and combined treatment. Therefore, for multidisciplinary teams, such as thoracic tumor board participants, the involvement of both neurosurgical colleagues and radiation oncologists is advisable, in order to develop the best treatment strategy for the patients with oligometastatic NSCLC lung cancer to the brain.

De Ruysscher and colleagues published a single-arm phase II trial (in 2012) describing the outcomes of patients with synchronous oligometastatic NSCLC [15]. These patients underwent local consolidative therapy (LCT), consisting of either surgery or radiotherapy, after they received systemic therapy. For this study, the authors defined the inclusion criteria for oligometastatic NSCLC as <5 metastases. Of note, patients with progressive disease while on systemic therapy were not excluded from LCT. Forty patients met the inclusion criteria. The primary tumor and its regional lymph nodes were treated with radiotherapy or chemoradiation, and no patients underwent surgical resection of this tumor or nodal disease. Nine patients were treated with surgical resection of their solitary metastasis. The median OS was 13.5 months. Median progression-free survival (PFS) was 12.1 months. De Ruysscher and colleagues published their long-term outcomes in a 2019 update, in which the seven-year results were reported [16]. The patients had an eight percent five-year PFS. Despite these relatively disappointing results, this report was the first prospective study on the PFS and OS that went beyond five years in NSCLC patients with oligometastatic disease. Based on this outcome, De Ruysscher postulated (in 2012) that careful patient selection and identification of “specific genetic characteristics that underlie the oligometastatic feature” would be necessary to bring the most benefit with local ablative therapies [14]. Given the inclusion criteria that allowed for the progression of systemic therapy and primary tumors that were not resected, this group of patients may not be the optimal group to study or manage the resection of oligometastatic disease.

In 2016, Gomez and colleagues published the first prospective multicenter dual-arm, randomized phase II trial, comparing systemic therapy alone to systemic therapy followed by LCT [17]. LCT and LAT (locally-aggressive therapy) are relative synonyms. The LCT in this report included either surgery or radiation, in addition to a combination of both for patients with NSCLC oligometastatic disease. The authors defined the inclusion criteria for oligometastatic disease as three or fewer metastases. The thoracic surgeon, Dr. Jessica Donnington, noted in her 2019 commentary “Keeping surgery relevant in oligometastatic non-small cell lung cancer” that the key to the success of the “Oligomez” trial was in the trial design [18]. The authors established that patients who had progressive disease in maintenance therapy were excluded from the trial. This exclusion criteria selected for patients with a less aggressive biology, but also decreased the number of patients eligible for LCT, which is a significant difference, compared to the phase II study conducted by De Ruysscher and colleagues [14]. With the change in patient selection, Gomez and colleagues fulfilled the challenge set by De Ruysscher to select for the “genetic characteristics that underlie the oligometastatic feature” [14]. With a median follow-up of 12.39 months, the LAT for all disease sites has a PFS of 11.9 months versus 3.9 months in the systemic therapy arm. The one-year PFS was 48% for patients undergoing LCT, compared to 20% for the patients receiving systemic therapy. Based on this data, the trial was closed early at the pre-specified interim analysis, based on the recommendations from the Data Safety Monitoring Committee.

In 2019, Gomez and colleagues published a subsequent report with a longer median follow-up of 38.8 months [19]. They observed a PFS of 14.2 months for the patients who underwent LCT versus 4.4 months for those on systemic therapy alone. Additionally, the OS in the LCT group was 37.6 months versus 9.4 months for the patients who received systemic therapy alone. This trial represents the landmark study for the evaluation of the benefit of LCT for the management of oligometastatic NSCLC. The study has several strengths. First, given that the role of LCT in oligometastatic disease was truly unknown, randomization to LCT versus systemic therapy-maintained equipoise, which enabled enrollment without significant bias toward one arm. Second, this trial accrued enough participants to reach statistical significance, unlike the prior attempts to address this question, which have been closed prior to completion, due to poor accrual. Despite the success of this trial, in terms of both accrual and results, only 49 participants were enrolled, and heterogeneity in the patients’ tumor did exist; therefore, this trial has not led to widespread adoption of LCT for oligometastatic disease in NSCLC.

Of note, the study highlighted that a response to systemic therapy (or, at least, the lack of disease progression while on therapy) is a good prognostic factor for outcome after LCT. Second, only 24% of patients in the study had surgery for LCT, whereas the remainder received radiotherapy. The patients who received radiotherapy were split between external beam and stereotactic body radiotherapy. The rationale for receiving radiotherapy over surgery is unknown and likely related to the bias of the treating clinicians toward less invasive radiotherapy over the potential risks associated with surgical resection.

In a separate report, Iyengar and colleagues published similar results to Gomez and colleagues in a randomized, controlled phase II clinical trial [20]. In this trial, both the primary site and sites of oligometastatic lesions were treated. Maintenance chemotherapy alone was compared to maintenance chemotherapy with radiation. The primary endpoint was PFS. Patients in this study did not undergo surgical resection. Twenty-nine patients were enrolled in this study. Fourteen underwent radiotherapy with chemotherapy (radio and chemo), and fifteen underwent chemotherapy alone. The authors observed a significant difference in PFS of 9.7 months in the radio and chemo arm versus 3.5 months in the chemotherapy alone arm. Similar to the study by Gomez and colleagues, the trial was stopped for accrual after 80% of the planned enrollment was achieved at interim analysis, due to the significant survival benefit in the radio and chemo arm. Interestingly, the patients in the radio and chemo arm had fewer distant recurrences. At the time of analysis, 10 of 15 patients in the chemotherapy arm had progressive disease; among those 10 patients, 7 had progression at the original site of disease. In the radio and chemo group, 4 of 14 had progression, but none had progression at the site of disease.

SABR-COMET, published by Palma and colleagues in 2019, with long-term results in 2020, is another landmark clinical trial in the discussion of the management of oligometastatic disease [21]. In contrast to the previously discussed trials, SABR-COMET included patients with primary cancers besides NSCLC. This phase II randomized study enrolled 99 patients, from 10 centers in 4 countries, with breast (*n* = 18), lung (*n* = 18), colorectal (*n* = 18), prostate (*n* = 16), or other (*n* = 29) primary tumors. Participants had 1–5 metastatic locations and were randomized to palliative standard of care (SOC) treatment or SOC plus stereotactic ablative radiotherapy (SABR), with a 51-month median follow-up. The primary outcome event, defined as death as a result of any cause, occurred in 73% of patients in the SOC arm and 53% of patients in the SOC plus SABR arm. The median PFS in the SOC arm was 5.4 months, and it was 11.6 months in the SOC plus SABR arm. This study adds to the growing body of evidence supporting the use of ablative therapies for oligometastatic cancers.

The Gomez, Iyengar, and Palma trials supported the use of local therapy for oligometastatic disease for local control, as well as OS. The Gomez and Iyengar studies both stopped accrual because of statistically significant survival advantages in the LCT interventional arm. However, none of these clinical trials characterized the distinction between a radio and chemo arm and surgery and chemo arm. No patients in Iyengar’s or Palma’s LCT arm and only 24% in Gomez’ LCT arm had surgery. So, the current data from the randomized controlled trials argues on behalf of LCT, in addition to systemic therapy, for oligometastatic disease; however, it does not clarify which local modality is superior. Clearly defining the distinction between surgery and radiotherapy, as treatment adjuncts, will require additional randomized trials.

### 1.3. Meta-Analyses

Several meta-analyses have evaluated the role of LAT for oligometastases in NSCLC. These studies have helped clarify patient selection, even though they cannot establish a causative association between LAT and survival.

Ashworth and colleagues reported a meta-analysis of 18 reports and 2 abstracts for a total of 757 patients with oligometastatic NSCLC [5]. The inclusion criteria consisted of patients with 1 to 5 metastases, with either synchronous or metachronous lesions. The primary tumor had to be controlled with surgery, radiotherapy, or a combination of both, regardless of use of systemic therapy. Eighty-three percent had surgery to manage the primary lesion. All sites of disease were targeted by local therapy, in which 62.3% underwent surgery, and 52.4% of the total underwent surgery alone without radiotherapy. The remaining 37.7% underwent LAT with radiotherapy only. The authors reported that adenocarcinoma histology, metachronous versus synchronous metastatic lesions, and the presence of intrathoracic lymph nodes were associated with long-term survival. The authors established risk groups based on recursive partitioning analysis (RPA), which divided patients into low-, medium-, and high-risk categories, based on these factors. The low-risk group with metachronous disease and no lymph node involvement had a five-year survival rate of 47.8%, whereas the patients in the high-risk group, defined by synchronous disease and nodal metastases, had a five-year survival rate of 13.8%. Ashworth and colleagues also highlighted that patients with oligometastatic NSCLC should be considered a distinct category that is separate from stage IV NSCLC, given that the five-year OS for the entire cohort was 29.4%, which is much better than the outcomes of the majority of patients with metastatic NSCLC.

In a separate report, Li and colleagues performed a meta-analysis in patients with synchronous oligometastatic NSCLC [22]. Their aim was to compare patients treated with ‘aggressive thoracic therapy (ATT)’ to those treated with medical management only. ATT was defined as surgery, radiotherapy, or a combination of both modalities [16]. This meta-analysis included seven observational, cohort studies, which reported on a total of 668 patients, ranging from 33–213 patients per report. They specifically evaluated patients with synchronous oligometastatic NSCLC and excluded metachronous lesions. They reported that ATT was associated with longer survival based on one-, two-, three-, and four-year survivals of 74.9%, 52.1%, 23.0%, and 12.6%, respectively, in the ATT group versus 32.3%, 13.7%, 3.7%, and 2.0%, respectively, for the patients treated medically only. Among the entire cohort, 277 patients (34%) were treated with ATT [16]. These numbers collectively suggest that ATT was associated with a 52% reduction in the risk of death [16].

While the results of these meta-analyses suggest a benefit for LAT, the selection bias of patients who underwent this approach is largely unknown. Therefore, despite the statistical benefit noted in these two reports, the patient cohorts are not randomized; so, the benefit is still unclear. However, the meta-analyses do provide survival estimates and prognostic factors, which can help with management decisions and counseling patients. Additionally, the survivals of these patients were significantly better than most reports for patients treated with systemic therapy alone. Therefore, the resection of oligometastatic disease in selected patients may be considered.

### 1.4. Retrospective Studies

Opitz and colleagues specifically evaluated patients with synchronous metastatic disease [23]. They performed a retrospective review of patients who underwent surgery of the primary tumor, with concurrent management of synchronous metastatic disease. This report included 124 patients from four centers. They defined oligometastatic disease as five or fewer synchronous metastases in less than or equal to two organs. All patients underwent surgery for their primary tumors. The metastatic lesions were resected in 72% of cases, and about half had radiotherapy, in addition to surgery. The remaining 28% received radiotherapy only. The most common locations of the metastatic disease were brain (61.3%), adrenal (10.4%), bone (9.7%), and lung (6.3%). A single metastatic lesion was present in 77.4%, and single organ involvement occurred in 97.6%. A total of 80% of patients survived one year, with 58% surviving two years and 36% surviving five years. The authors found that lymph node involvement, bone metastases, and age over 60 were negatively associated with both two- and five-year OS [21]. The median OS, with or without nodal disease, was 20 months versus 78 months, respectively. The association with age of 60 correlated with OS but not PFS.

Similar to Optiz and colleagues, Mordant and colleagues performed a retrospective review of 94 patients with synchronous, non-thoracic M1b metastatic disease [24]. This study differed from the report by Optiz and colleagues, in that the lung metastases were not included, which accounted for only 6.3%. On both univariable and multivariable analyses, the authors noted that histology, nodal involvement, tumor size, and induction therapy were significant prognostic factors. The median OS, with or without nodal disease, was 9 months versus 25 months, respectively. Interestingly, these numbers are quite different than the 20 months and 78 months results noted by Optiz and colleagues, which suggests that significant clinical selection bias exists for patients who undergo these resections [21]. This study added that, for patients who received induction therapy, the median OS was 35 months, compared to 10 months for patients who did not receive therapy. While this finding may suggest that induction therapy prior to these resections is indicated, it may represent another selection criterion, based on exclusion of patients who have progressive disease or whose performance status deteriorates during therapy. Ideally, a prospective, intent-to-treat design would report the number of patients who do not proceed to therapy. As another prognostic factor, they reported that pneumonectomy had a worse outcome, compared to lobectomy. Interestingly, they observed that surgical resection versus other methods for management of the metastases was not a prognostic factor. Based on these findings, the authors suggested that metastatectomy for a solitary oligometastatic lesion may only be indicated in symptomatic lesions. They concluded that these lesions are “low grade” tumors and can be controlled by radiotherapy or systemic chemotherapy. Yet, as the authors note, this study was not powered as a non-inferiority study; therefore, they may have overlooked a difference when one exists, making it difficult to derive a strong conclusion from this study.

In addition to the surgical management of oligometastatic disease, a related (but different) topic is the management of the primary thoracic lesion in patients with oligometastatic NSCLC. Mitchell and colleagues performed a retrospective study of 88 patients who underwent LCT of the primary tumor at a single institution (from 2000–2017) [25]. All patients had oligometastatic NSCLC, which they defined as ≤3 lesions. Similar to the studies above, they specifically evaluated patients with synchronous lesions. The most common sites of oligometastatic disease were brain (53.4%), bone (26.1%), or adrenal (11.4%), which is relatively consistent among most of these studies. The authors specifically queried the outcomes of the patients after management of the primary tumor, as opposed to LCT of the metastatic disease. The primary lesion was treated with surgery or radiotherapy in 28.4% or 71.6%, respectively. The median OS for surgery and radiotherapy were 55.2 months versus 23.4 months, respectively. These results were not directly compared; therefore, statistical significance was not evaluated, even though the outcome with surgery appears longer. In contrast, they did evaluate the differences in the sites of first failure, cumulative incidence of locoregional failure, and systemic progression. The locoregional treatment failure was evaluated, which revealed one- and three-year freedom from locoregional progression of 87.7% and 76.9% for radiotherapy and 91.1% and 86.5% for surgery. Most recurrences were systemic, in which 52.4% in the radiotherapy group and 41.1% in the surgical group experienced systemic progression. Based on these reasonable outcomes, they concluded that resection of the primary tumor can be considered in well-selected patients; however, they have planned a phase III study to hopefully answer whether this approach is beneficial.

Similar to the study by Mitchell and colleagues, Yang and colleagues addressed the question of management of the primary lesion [26]. The authors performed a retrospective review of the National Cancer Database (NCDB), which evaluated patients with stage IV NSCLC from 2004 to 2013. Among these patients, they identified patients in which the primary tumor was treated with surgery. Three thousand and ninety-eight patient records were included. As with other reports, they noted that increased tumor size and positive nodal disease were associated with worse survival. Additionally, the authors recommended avoiding pneumonectomy, which is consistently associated with worse outcomes. The long-term outcomes of the patients treated with surgery were compared to patients treated with systemic therapy. They reported that surgical resection, in addition to other treatment options, was associated with five-year OS of 25% for the subset of patients with cT1–2, N0–1, M1 or cT3, N0, or M1 NSCLC. The outcomes of these patients were significantly better than those treated with chemotherapy, chemoradiotherapy, or radiation alone. However, the authors do note that these comparisons are subject to selection bias, and the overall demographics of the patients are not the same, which may account for differences in survival. However, as with other retrospective reviews, tumor size and lymph node status were meaningful prognostic indicators in a group of patients whose outcomes are reasonable for metastatic NSCLC [21,22,27,28]. Additionally, the survival rates are quite reasonable, so resection can be considered in the appropriately selected patient.

### 1.5. Current NCCN Guidelines

Current NCCN guidelines recommend only therapeutic intent surgical treatment for patients with stage IIIA disease who have single station nodal disease or stage IV patients with oligometastatic disease to the brain or adrenal glands. Most of the surgical management options are for patients who do not fit the guidelines well.

## 2. Conclusions

Patients with oligometastatic NSCLC are a distinct subgroup, compared to patient with widely metastatic NSCLC. Emerging evidence suggests that LAT for oligometastatic disease may be a reasonable option in select patients. Based on retrospective reviews, as well as the meta-analyses, increase in the T status, presence of nodal metastases, the number of lesions, and performing a pneumonectomy versus lesser resections were associated with worse outcomes. In addition to these studies, the randomized reports with LAT with surgery or radiotherapy alone were stopped for accrual, secondary to achieving statistically significant improvement in survivals in the LAT arms. Additionally, the survival rates of these patients were significantly better than most reports for patients treated with systemic therapy alone. Therefore, the resection of oligometastatic disease in select patients may be considered. Given the complexity of the decisions regarding resection of the primary tumor, as well as the metastatic lesions, discussion at tumor boards with radiation oncologists, oncologists, pathologists, thoracic surgeons, and, occasionally, neurosurgical colleagues can help define the best treatment strategy.

## Data Availability

All data reported is contained within the attached references.

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
