# Peer review of "The Role of Surgery for Oligometastatic Non-Small Cell Lung Cancer"

_cancers, 2022, doi:10.3390/cancers14102524_

Round 1
Reviewer 1 Report
The authors introduced the concept, Oligometastatic non-small cell lung cancer (NSCLC), which refers to a limited number of metastatic sites. By the evidence shown in the reference, they concluded that oligometastatic NSCLC should be considered as a subgroup from other widely metastatic NSCLC. It was summarized in the manuscript why oligometastatic NSLCL is special and the treatment options for this subgroup were suggested as well.
I find the manuscript was written professionally and detail-orientated. I suggest accepting the manuscript.
Author Response
I appreciate this reviewer's evaluation of this paper. Thank you for the feedback.
Reviewer 2 Report
This is an interesting review on a poorly treated issue. So I am in favor of this review. However English must be heavily addressed (typos, grammar and phrasing). It is clearly translated from another language to English and the structure is odd.
Minor:
line 110 : please mention the % of superior treatments related to reference 12
line 143: LCT should not be acronymized again
line 189: For clarity I would rather call the radio arm radio + chemo arm
lines 201-202: elaborate on their statement related to surgery, while only discussing radio and chemo, also considering that at line 207 they mention surgery and radio. They should better the logic of these paragraphs in a more consistent manner.
lines: 257-258 not clear what 80, 58 and 36 are referring to
line 315: T status is too colloquial
Author Response
This is an interesting review on a poorly treated issue. So I am in favor of this review. However English must be heavily addressed (typos, grammar and phrasing). It is clearly translated from another language to English and the structure is odd.
I appreciate this reviewer’s feedback. English is our first language. I have performed syntax and grammatical revision to better convey English nativity throughout the paper. Please see the attached document for the full paper with red-lined revisions.
Minor:
line 110 : please mention the % of superior treatments related to reference 12
In order to quantify the improved survival found in Vecht’s study with WBTR with surgical resection compared to WBTR alone I have included a reference to the improved median survival. The sentence now reads as follows:
“For patients with single metastasis to the brain and limited systemic disease, a randomized trial by Vecht and colleagues demonstrated median survival of 7 months with whole brain radiotherapy (WBTR) and 12 months with WBRT combined with surgical resection.12”
line 143: LCT should not be acronymized again
I made this revision in the body of the text.
line 189: For clarity I would rather call the radio arm radio + chemo arm
lines 201-202: elaborate on their statement related to surgery, while only discussing radio and chemo, also considering that at line 207 they mention surgery and radio. They should better the logic of these paragraphs in a more consistent manner.
I have reworked this section:
“Twenty-nine patients were enrolled in this study. Fourteen underwent radiotherapy with chemotherapy (radio + chemo), and 15 underwent chemotherapy alone. The authors observed a significant difference in PFS of 9.7 months in the radio + chemo arm versus 3.5 months in the chemotherapy alone arm. Similar to the study by Gomez and colleagues, the trial was stopped for accrual after 80% of the planned enrollment was achieved at interim analysis due to the significant survival benefit in the radio + chemo arm. Interestingly, the patients in the radio + chemo arm had fewer distant recurrences. At the time of analysis, 10 of 15 patients in the chemotherapy arm had progressive disease and among those 10 patients, 7 had progression at the original site of disease. In the radio + chemo group, 4 of 14 had progression, but none had progression at the site of disease.”
lines: 257-258 not clear what 80, 58 and 36 are referring to
I have revised this sentence as follows:
“80% of patients survived to one year, 58% to two years, and 36% to five years.”
line 315: T status is too colloquial
I have revised this sentence as follows:
“As with other reports, they noted that increased tumor size and positive nodal disease were associated with worse

Reviewer 3 Report
The manuscript is a review of local aggressive therapies for oligometastatic NSCLC. The paper covers some major differences in synchronous v. metachronous, location of mets, nodal v. no-nodal disease. The authors highlight which studies included surgery versus stereotactic radiation, or both. They include prospective studies and meta-analyses.
Abstract 13-15: Sentence needs clarification
Intro: 33: Crizotinib example is vastly outdated. (ALEX, ALTA-1L and CROWN represent the current first-line agents. All have PFS in the range of 2-3 years.And your reference is for ALEX) New targeted therapies for for EGFR, ALK, ROS1 offer years of added survival. MET exon 14 skipping, RET fusion, KRAS G12C and BRAF likely do no offer the same degree of benefit - but arguably the efficacy of these TKIs support more aggressive local treatment of oligo/poly-metastatic disease. I think it is worth adding a sentence about how improved outcomes with systemic therapy may allow a larger group of patients the benefit of LAT.
129 - is that intended to say, "none" instead of nine?
197- I think SABR-COMET, Palma et al. should be added. It allows multiple sites (only 20% NSCLC) but again I think it supports LAT to 1-5(or 1-3) metachronous lesions. I think SABR-COMET and Gomez are the two most relevant studies in oligometastatic NSCLC
249 - "patients"
Overall a good review - the strengths are how well it defines number of mets, synchronous v metachronous and some major prognostic factors.
Author Response
Abstract 13-15: Sentence needs clarification
I have reworked the abstract (and the whole paper) for improved clarity, grammar, and syntax. Please see the attached document with red-line edits.
Intro: 33: Crizotinib example is vastly outdated. (ALEX, ALTA-1L and CROWN represent the current first-line agents. All have PFS in the range of 2-3 years.And your reference is for ALEX) New targeted therapies for for EGFR, ALK, ROS1 offer years of added survival. MET exon 14 skipping, RET fusion, KRAS G12C and BRAF likely do no offer the same degree of benefit - but arguably the efficacy of these TKIs support more aggressive local treatment of oligo/poly-metastatic disease. I think it is worth adding a sentence about how improved outcomes with systemic therapy may allow a larger group of patients the benefit of LAT.
I appreciate this feedback. I have included reference to second generation TKI’s, their improved effectiveness, and the expanded opportunity for LAT that comes with more effective targeted therapy:
"Platinum based chemotherapy extends survival from early reports of 3-4 months to 12 months. First generation targeted therapy offers similar results.5Patients with advanced NSCLC harboring rearrangement of the anaplastic lymphoma kinase (ALK) gene can receive crizotinib,6and median progression free survival (PFS) with this drug is only 10.9 months. Alectinib, a second generation ALK inhibitor was found to be more efficacious, extending PFS to 34.8 months.(Editor--Please insert Mok reference) Other first line agents target Epidermal Growth Factor Receptor (EGFR) and ROS proto-oncogene 1 (ROS1). All of these modern targeted therapies offer a PFS between two and three years. There is now growing optimism that improved targeted therapies will allow a larger cohort of patients the chance to benefit from locally aggressive treatment (LAT) with surgery or radiotherapy. But, in spite of these advances in systemic options, truly long-term survival is still relatively rare. Additionally, the management of patients with oligometastatic disease versus metastatic disease is less defined."
129 - is that intended to say, "none" instead of nine?
It is meant to say “nine.” For clarity I have revised this sentence to read as follows: “nine patients were treated with surgical resection of their solitary metastasis.”
197- I think SABR-COMET, Palma et al. should be added. It allows multiple sites (only 20% NSCLC) but again I think it supports LAT to 1-5(or 1-3) metachronous lesions. I think SABR-COMET and Gomez are the two most relevant studies in oligometastatic NSCLC
I have included the following paragraph describing the SABR-COMET trial:
“SABR-COMET, published by Palma and colleagues in 2019 with long-term results in 2020, is another landmark clinical trial in the discussion of management of oligometastatic disease.(Editor—please insert reference to Palma trial). In contrast to the previously discussed trials, SABR-COMET included patients with primary cancers besides NSCLC. This phase II randomized study enrolled 99 patients from 10 centers in 4 countries with breast (n = 18), lung (n = 18), colorectal (n = 18), prostate (n = 16) or other (n = 29) primary tumors. Participants had 1-5 metastatic locations and were randomized to palliative standard of care (SOC) treatment or SOC plus stereotactic ablative radiotherapy (SABR) with a 51-month median follow-up. The primary outcome event , defined as death as a result of any cause, occurred in 73% of patients in the SOC arm and 53% of patients in the SOC plus SABR arm. Median PFS in the SOC arm was 5.4 months and 11.6 months in the SOC plus SABR arm. This study adds to the growing body of evidence supporting the use of ablative therapies for oligometastatic cancers.”
249 - "patients"
I have made this revision.
